# An HS-SPME-GC-MS Method for Profiling Volatile Compounds as Related to Technology Used in Cider Production

**DOI:** 10.3390/molecules24112117

**Published:** 2019-06-04

**Authors:** Jakub Nešpor, Marcel Karabín, Kateřina Štulíková, Pavel Dostálek

**Affiliations:** Department of Biotechnology, Faculty of Food and Biochemical Technology, University of Chemistry and Technology, Prague, Technická 5, 166 28 Prague 6-Dejvice, Czech Republic; Jakub.Nespor@vscht.cz (J.N.); Marcel.Karabin@vscht.cz (M.K.); Katerina.Stulikova@vscht.cz (K.Š.)

**Keywords:** HS-SPME-GC-MS, cider, volatile compounds

## Abstract

Volatile aromatic compounds have a major impact on the final organoleptic properties of cider, and their profiles are influenced by a number of parameters that are closely related to production technologies, especially with regard to the raw material, the microorganism used and the fermentation process. In this work, the profiles of volatile compounds (4 fatty acids, 6 higher alcohols and 12 esters) from 34 European ciders were studied using HS-SPME-GC-MS. Volatiles were isolated by HS-SPME using a CAR/PDMS fiber. Analytical data were statistically evaluated using principal component analysis, and differences in composition of volatiles between cider produced using “intensified” technologies and that of ciders produced by traditional production processes were demonstrated. This difference was mainly due to varying contents of some higher alcohols (2-methylpropanol, isoamyl alcohols, hexanol, and butane-2,3-diol) and esters (ethyl 2-methylbutanoate, butylacetate, and hexyl acetate).

## 1. Introduction

Cider, a traditional alcoholic beverage with a long production history, is produced in several countries on all continents except for Antarctica. Great Britain is currently the top European producer, followed by France and Spain, but cider production has increased dramatically in other countries, particularly the USA, Germany, Canada, Brazil, Argentina, Chile and Paraguay [1]. Nowadays, cider is becoming more and more popular, and one example of such popularity can be found in the increasing consumption in countries without a long tradition of cider-making, such as the Czech Republic, where the rate of growth of the cider market is one of the highest in Europe [2]. Popularity may be connected with the specific organoleptic profiles of ciders, which are closely associated with the variable contents of many aromatic compounds in the final beverage. One such group is commonly known as “volatile compounds”, covering a number of different classes of chemical compounds—alcohols [3,4], short and medium chain fatty acids and esters [5,6]. Some of these components originate from apples, but most of them are formed during fermentation and maturation [7]. The most important alcohols in cider include 2-phenylethanol [8], propanol, butanol, hexanol, 2-methyl-propanol, 2-methyl-butanol, 3-methyl-butanol, 2,3-butanediol and glycerol [9]. These substances have a typically warm or sweet sensory character and 2-methylbutanol and 3-methylbutanol also give cider an alcoholic flavor [10]. Esters with a significant impact on the final sensory profile are hexyl acetate with an aroma of green apples [11], ethyl-2-methylbutanoate with an aroma of yellow apples [12], isoamyl acetate, which is characterized by its banana flavor [11] and also ethyl lactate, ethyl butanoate, ethyl hexanoate, ethyl octanoate and diethyl succinate [13]. The organic acids represented in the highest concentrations are formic, acetic, lactic, citric, fumaric, pyruvic and malic acids [14]. These acids generally give ciders a sour or fresh taste and citric acid is known for its lemony aroma [14].

Besides parameters related to the quality of the raw material (variety and maturity of apples, growth conditions), the intensity and character of the cider flavor is influenced in particular by the technology used [15,16,17]. Traditional production of cider resembles, in many respects, the production of sparkling wines [18]. From a technological point of view, it can be divided into two steps. The first stage is the production of fresh apple must (usually from local apples), and the second consists of fermentation and maturation [17], supplemented by post-fermentation adjustments, such as filtration and pasteurization [7,18]. Industrial large-scale production usually includes deep attenuation of the apple-juice concentrate, or blending of a fully fermented and dearomatized apple wine [19] with apple juice or aromatized apple concentrate. 

All these aspects are based on local customs and legislative regulations. For example in France, a country in which traditional cider making methods are well-respected, the use of sugar to increase the sugar content of the must or for sweetening the finished cider is completely banned. Traditional French ciders are produced in small volumes, especially in the regions of Normandy and Brittany [8], and are characterized by an astringent taste (due to the high content of tannins), apple flavor and low alcohol content [1,9]. The strictness of such an approach to cider-making in francophone areas is documented by the protected designation of origin based on “The appellation d’origine contrôlée—AOC”, used for ciders produced from specific apples by traditional procedures [20]. 

In contrast, in the Czech Republic, where there is no historical tradition of cider production and cider is a quite a new product on local markets, legislation is much more relaxed and permits products to be labeled as cider that are prepared by fermentation of not only fresh must, but also apple juice concentrate or dried apple juice that has been rehydrated with water, or even apple wine diluted with apple juice. Fresh must, sugar, CO_2_ or apple juice concentrate can be legally added after fermentation and maturation [19].

Considering the chemical properties and concentrations of the volatile compounds involved, gas chromatography combined with mass spectrometry (GC-MS) seems to be the best technique for identification and quantification [6,12,21,22]. However, a proper isolation and concentration technique should be applied before the chromatographic analysis due to the presence of many other cider components, such as sugars, which can cause serious damage to the chromatographic system [10]. 

So far, the influence of the technology on the composition of volatiles has not been statistically confirmed in any previous study. Therefore the aims of this study were to analyze the contents of volatile compounds, as determined by HS-SPME-GC-MS, in European ciders, and to identify and statistically confirm the impact of fermentation technologies on the chemical composition that defines the organoleptic properties of the product generally. 

## 2. Results and Discussion

### 2.1. Variability of Volatiles in the Cider

In ciders, as well as in similar fermented alcoholic beverages, ethanol is the major volatile constituent arising from fermentation, followed by higher alcohols, esters and short and medium chain fatty acids [22,23]. In the samples analyzed, the content of higher alcohols ranged from 13.4 to 86.9 mg/L, esters from 7.0 to 56.6 mg/L and short and medium chain fatty acids from 3.0 to 41.2 mg/L (Table 1).

The majority of ciders analyzed contained higher levels of alcohols than esters, which is probably due to the fact that the major quantity of higher alcohols is produced during the primary phase of fermentation as a result of intense metabolic transformations of proteins that were necessary for biomass growth during primary production [24]. On the other hand, the formation of esters, mostly by direct esterification of acids with these alcohols, takes place subsequently, mainly during maturation. Up to 75% of all higher alcohols are generated by amino acid deamination and subsequent reduction of their oxo-forms, 10% of higher alcohols are formed as a by-product of sugar utilization and the remaining 15% through non-specific biochemical pathways [24]. The rate of growth of biomass is significantly affected by dissolved oxygen, so increased aeration at the beginning of fermentation leads to increased concentrations of higher alcohols [24]. The importance of amino acid metabolism for the formation of higher alcohols is confirmed by the fact that alcohols that reached the highest concentration were amyl alcohols (2-methyl- and 3-methylbutanol) as well as another branched alcohol, 2-methylpropanol, which are predominantly formed from corresponding amino acids leucine, isoleucine and valine, respectively [24]. 

A higher concentration of hexanol, which is one of the main precursors of the typical apple flavor, was observed especially in ciders made from fresh unpasteurized must, because heat treatment (pasteurization) leads to the release of volatiles from the must so their levels in finished products are lower [25].

A higher concentration of butane-2,3-diol in fermented beverages is the result of reduction of diacetyl, mainly by bacterial microflora [26]. Therefore the highest concentrations of butane-2,3-diol were found in samples 23 and 24 (concentration 41.5 and 44.6 mg/L respectively), which are very specific samples made from sour apple varieties using exclusively native microflora (yeast and bacteria) from the surface of apples; this enables high attenuation of the must. Samples 19, 21, 22 also contained higher concentrations of butane-2,3-diol than the majority of samples. These ciders were fermented using a combination of natural microflora and a pure yeast culture, which partially suppressed the growth of wild-type microorganisms [14]. Therefore, the relative amount of butane-2,3-diol was not so high in these samples in comparison with samples 23 and 24. Another explanation may be based on differences in fermentation and maturation parameters, manifested in lower concentrations of ethanol in samples 19, 21, 22. Concentrations of butane-2,3-diol over the above were also detected in samples 1, 4 and 5. All of these samples were industrially produced in large volumes from juice apple concentrate using a pure yeast culture, which leads to the elimination of the influence of wild cultures originating from apples. Therefore, the increased concentrations of these compounds are probably due to higher temperatures of fermentation, commonly used to reduce diacetyl, and to accelerate fermentation. These intensified processes may not be long enough for sufficient degradation of butane-2,3-diol [27], and thus these ciders contain high concentrations of butane-2,3-diol, but not as high as ciders fermented by wild or mixed cultures [28].

Esters are another important group of sensory-active compounds in alcoholic beverages, mainly as the source of fruity and/or floral aromas in finished products [5]. The esters are mainly formed by esterification of fatty acids with ethanol or higher alcohols arising from the Ehrlich pathway [29]. The most common esters in ciders are ethyl-esters, derived from ethanol, and acetates derived from acetic acid [13,30]. The esters that were mostly in concentrations close to their thresholds were isoamyl acetate (banana aroma), [22], 2-phenylethyl acetate (honey and floral aroma), [22] and hexyl acetate, which is produced from hexanol, an intermediate of fatty acid metabolism [31], and is known as one of the basic components of cider apple flavor [32].

When maturation is carried out through secondary fermentation in a bottle by a wild or mixed microflora (cider 24, 23, 13, 10 and 31), the level of esters, especially ethyl acetate [24], increased significantly as a result of prolonged malolactic fermentation [21], while the content of the majority of higher alcohols decreased [23]. One of the main factors determining the final ratio of higher alcohols to esters was the amount of fermentable nitrogenous substances in the must, and the maturation time [24,33].

The content of volatile fatty acids also varied depending on some technological parameters, as documented by a comparison of industrial and craft ciders. Apart from the different raw materials, the crucial difference between these groups was the maturation time. Intensified (faster and deeper) fermentation leads to limited formation of fatty acids as well as other sensory active volatiles. That was proven by the correlation between the amount of ethanol and the total content of aromatics (higher alcohols, esters and fatty acids). The correlation coefficient was 0.3922, which is sufficient to confirm a correlation with a level of significance *p* = 0.05 for 34 different samples. A similar correlation was published previously [34].

### 2.2. Influence of Production Technology

On the basis of the above, it can be hypothesized that individual technological factors (raw material, fermentation method, and type of microorganism) produce significant differences in the chemical composition of ciders, and this can be documented on the basis of the volatiles profile. Using this profile, it is possible to differentiate statistically between ciders produced on large-scale using intensified technological processes and ciders manufactured by small-scale producers preferring more traditional technologies. 

Principal component analysis and factor analysis were applied on two groups of different samples to confirm the assumption that the use of apple juice concentrate or blending of apple wine and apple juice, together with a different approaches to processing, will affect the contents of volatile substances and therefore these samples will have different organoleptic properties. Data are presented in the form of a scatterplot (Figure 1) and factor loading (Table 2) showing that differences in volatile profiles between the products from small craft manufacturers and large-scale producers led to a separation of samples based on values of the two most important factors, 1 and 2. Factor 1, describing 23.90% of the total variability, includes the contents of octanoic acid and three esters (isoamyl acetate, ethyl hexanoate and ethyl octanoate). Factor 2, defining 19.78% of variability, includes the ester ethyl-2-methylbutanoate and two branched chain alcohols (2-methylpropanol and 2-methylbutanol). In addition, statistically significant differences in concentrations of butane-2,3-diol (factor 3) were found.

These statistically significant dependencies indicate that differences in the composition of cider volatiles are attributable partially to compounds already occurring in higher concentrations in the raw materials (ethyl 2-methylbutanoate) [33], but also to compounds such as isoamyl acetate and branched alcohols formed during fermentation, whose contents may be affected by the fermentation conditions (temperature, time, and microflora) [7,17]. 

The lower content of volatile substances in ciders produced in large scale can be caused, for example, by the use of different raw materials (apple juice concentrate or apple wine) that usually contain fewer volatile compounds and their precursors, e.g., free amino acids that influence the composition of higher alcohols [24]. An intensified production process using rapid fermentation without maturation or aging can also affect the volatiles profile [25], creating or reducing volatile compounds, especially esters [5,35]. This was confirmed by similarities between sample 25 from a large-scale producer and ciders from small-scale manufacturers. Sample 25 contained a higher concentration of esters as a result of maturation time, and samples 14, 26 and 30 from small-scale producers were more similar to products from large cider houses, as they were not being intensively fermented and contained lower levels of ethanol. As indicated by the alcohol content of samples 14, 26, 30, a shorter fermentation time also leads to lower levels of secondary fermentation products such as higher alcohols and especially esters [36]. The ratio of higher alcohols to esters was therefore more similar to ciders in the red group (Figure 1). 

The impact of using different raw materials and unusual techniques was demonstrated in the group of samples 2, 4, 5, 7, which clustered separately from the rest of the ciders from large-scale production, these ciders being made by mixing de-aromatized wine with fortified apple concentrate. Such a production format also affected the volatiles composition and these ciders had very distinct organoleptic properties.

An important technological aspect affecting fermentation is pressure in the fermentation vessel. If cider is produced in large volumes, the hydrostatic pressure is higher and negatively affects the formation of volatile aromatic compounds [24,37]. Higher pressure also increases the concentration of dissolved carbon dioxide, which leads to a decrease in the uptake of free amino acids, the precursors of sensory active substances, and consequently to a reduction in the production of higher alcohols and acetates [38].

Another effect leading to a different profile of volatiles is a different post-fermentation approach. Modern kieselguhr-filtration, usually used for the industrial filtration of beer, wine and cider, is generally considering as very gentle [39]. However, it was found that during filtration, the content of higher alcohols, as well as the content of volatile and non-volatile acids, decreased, even though the ester content did not change significantly [39]. This suggests that esterification is not the main factor [30], and because filtration only takes a few hours, this is not sufficient for a significant change in the composition of the volatile substances [30]. The origin of these changes is probably related to the adsorption of volatile substances to kieselguhr [39], whose higher affinity for alcohols than for esters has already been demonstrated [22,39].

Differences in the composition of volatile substances in the ciders analyzed can also be explained by pasteurization, which leads to a decrease in the content of higher alcohols, esters and fatty acids [40].

### 2.3. Influence of Pitching Microorganism

One of the most important factors affecting fermentation is the microorganism used. For some of our samples (16 from 34 ciders, all from small-scale producers using fresh apple must), the general composition of the inoculum and place of origin were known. It was either pure yeast *Saccharomyces cerevisiae*, a mixture of yeasts and bacteria from the surface of apples (autochthonous culture) or an autochthonous culture with the addition of pure yeast (Table 4). The aim of this comparison via principal component analysis was to find statistically significant differences in the composition of these samples and to explain whether the choice of microorganism or the mixture influenced the profile of volatiles [17]. The results are presented in the form of a scatterplot (Figure 2) and show that samples could be divided into groups based on factor 1 (35.63% variability), which includes the content of the branched alcohol 3-methylbutanol, octanoic acid, and five esters (butyl acetate, isoamyl acetate, ethyl hexanoate, ethyl octanoate and ethyl decanoate), and factor 3 (15.58% variability) which includes (ethyl acetate, hexanol, acetic acid, 2-phenylethanol). In addition, statistically significant differences in concentrations of hexyl acetate (factor 2), and medium chain fatty acids (C-6, C-10), (factor 4) were found. Individual loadings for all compounds are shown in Table 3.

The dissimilarity of Spanish ciders 23 and 24 and one Czech cider 13, produced exclusively by autochthonous culture, was confirmed. These samples contained the highest levels of the majority of volatile substances, in particular acetic acid and ethyl acetate. Samples 23 and 34 contained the highest levels of butane-2,3-diol. The second group, consisted of ciders 19, 20, 21, 22, 26, 30—all French, not so deeply attenuated, which were produced from unpasteurized apple must containing wild microorganisms, but were pitched with strains of *Saccharomyces cerevisiae*. The last one was a group of English ciders, 14, 15, 16, 17, and 18 that were fermented exclusively using a pure yeast culture. These ciders contained predominantly higher alcohols because yeast metabolism is characterized by the production of higher alcohols rather than esters [28]. In this group was also one Czech cider, 10, which had the highest content of 2-phenylethanol from all samples. It was produced exclusively by autochthonous culture, but was not maturated for as long as samples 13, 23, and 24, and therefore the composition of volatiles was mainly higher alcohols, especially 2-phenylethanol, which is formed from phenylalanine via metabolic transformations under anaerobic conditions [41]. This alcohol occurs naturally in apple juice in trace amounts but its concentration increases rapidly during the main fermentation [42]. In addition, in sample 31, 3-methylbutanol was the second highest of all ciders and this sample also had a high content of acetic acid and ethyl acetate; it therefore stands at a distance from other samples in the scatterplot. 

## 3. Material and Method

### 3.1. Description of Samples

Thirty-four different ciders from seven different European countries (Czech Republic, Slovakia, France, Great Britain, Ireland, Spain and Estonia) were obtained from local markets. Eleven of them were manufactured by producers using modern, intensified technologies such as fast fermentation of pasteurized apple-juice concentrate (AJC) by a pure yeast culture (*Saccharomyces cerevisiae*) or by blending high alcohol content apple wine with apple juice in large-scale. The rest of the ciders were manufactured by smaller producers using classical techniques from unpasteurized must (UM), either exclusively using fermentation by wild microflora from apples, or by adding a pure yeast culture. An overview of the technological parameters for individual ciders is given in Table 4.

### 3.2. Reagents

All standards at analytical purities were obtained from Fluka (DE), Sigma-Aldrich (USA), and Alfa Aesar (DE). Gas chromatography-grade ethanol was purchased from Merck (DE). Sodium chloride was purchased from Penta (CZ). Demineralized water was prepared using a Milli-Q Millipore water purification system (Millipore, Bedford, MA, USA).

### 3.3. Ethanol Content Determination

Ethanol content in all ciders (Table 4) was determined by HPLC using an Agilent 1200 equipped with refractive index detector RID1 A (Agilent Technologies, Santa Clara, CA, USA). A chromatographic column WATREX 250 × 8 mm Polymer IEX H form, 8 µm (Agilent Technologies, Santa Clara, CA, USA) was used for separation, conditions of determination were set according to method published by Strejc, Siristova, Karabin, Almeida e Silva, and Branyik, (2013) [43]. All samples were analyzed at least in triplicate. 

### 3.4. Sample Preparation 

The basic sample preparation method and GC/MS were carried out in accordance with [44]. Before analysis, all ciders were stored at 4 °C, then 200 mL samples of these ciders were shaken for 7 min on a shaker (frequency of 175/min) to reduce the CO_2_ content. After shaking, ten mL of the sample together with 1 g of NaCl and 100 μL of an internal standard solution (ethyl heptanoate 11.68 g/L and 3-octanol 21.83 g/L in 70% (*v/v*) ethanol/water solution) were placed into a 20 mL dark vial sealed with a PTFE-silicone septum (Supelco; USA) and stirred on a vortex mixer until all salt was dissolved (1 min). Volatiles were isolated by head space–solid phase microextraction (HS-SPME) using an 85 μm Carboxen^®^/Polydimethylsiloxane (CAR/PDMS) SPME fiber. The isolation was performed for 40 min at 50 °C in agitator. Conditions of the extraction were set according to a method published by Nespor, Karabin, Hanko, and Dostalek, (2018) [44], with modification, such as different time of extraction for the specific matrix of the cider.

### 3.5. GC-MS Condition

GC-MS analyses were performed on an Agilent 6890N GC gas chromatograph (Agilent Technologies, Santa Clara, CA, USA) coupled with a single quadrupole Agilent 5975B Inert MSD mass detector (Agilent Technologies, Santa Clara, CA, USA). A polar DB-624 UI (30 m × 0.250 mm × 1.40 µm; Agilent Technologies, Santa Clara, CA, USA) column was used. The flow rate of carrier gas (Helium 6. 0—Linde Gas, DE) was 1 mL/min. The injection port was heated to 260 °C, desorption time was set at 10 min, and analyses were performed in 2:1 split mode. The temperature program of the column oven was started for 10 min at 30 °C followed by an increase to 52 °C with a 2 °C/min gradient. After a 2 min delay, the same gradient was used to increase the temperature to 65 °C, which was then held for 2 min. Finally, the temperature was increased at 5 °C/min to 250 °C and held for 3 min. The ionization energy was 70 eV and scanned mass range was 20-500 Da. 

### 3.6. Identification and Statistical Evaluation

Individual compounds were identified by their specific retention time and by comparing mass spectra with standard mass spectra listed in the National Institute of Standards and Technology MS 2.0 spectral database. Quantification of individual compounds was performed using four-point external calibration curves obtained by analyzing cider with the addition of different concentrations of standards. All analyses were performed in triplicate and peak areas were unified using internal standards (externally mixed solution of ethyl heptanoate and 3-octanol). The linearity of the calibration curves for each substance were verified from squares of the correlation coefficient (*r^2^*), which was not lower than 0.95.

Statistical evaluation was performed in Statistica software, version 12 (Statsoft, Inc., Tulsa, OK, USA). The results were statistically evaluated using a factor analysis (FA—Factor Analysis) and a principal component analysis (PCA—Principal Component Analysis), and processed into a scatter-plot.

## 4. Conclusions

In this work, a set of 34 European ciders was analyzed by HS-SPME/GC-MS. Fifty eighth compounds were identified and 22 of these compounds (4 short and medium chain fatty acids, 6 higher alcohols and 12 esters) were quantified by HS-SPME-GC-MS using a polar DB-624 UI GC-column and 85 μm Carboxen^®^/Polydimethylsiloxane (CAR/PDMS) SPME extraction fiber.

Principal component analysis confirmed the differences between ciders produced by craft and large-scale producers, arising as a result of using different technological approaches. The most significant differences, resulting from the use of different raw materials and production technologies, were in the differing compositions of higher alcohols (2-methylpropanol, 2-methylbutanol and butane-2,3-diol), esters (ethyl-2-methylbutanoate, isoamyl acetate, ethyl hexanoate and ethyl octanoate) and octanoic acid. Also, the influence of microorganisms (autochthonous culture, pure *Saccharomyces* yeast strains or a combination of both) was manifested in differing contents of 3-methylbutanol, 2-phenylethanol, hexanol, ethyl acetate, butyl acetate, isoamyl acetate, ethyl hexanoate, ethyl octanoate and ethyl decanoate, and two acids (acetic and octanoic). This suggests that traditional technologies, used in specific areas, are a prerequisite for preserving the characteristic sensory properties of local types of cider.

## Figures and Tables

**Figure 1 molecules-24-02117-f001:**
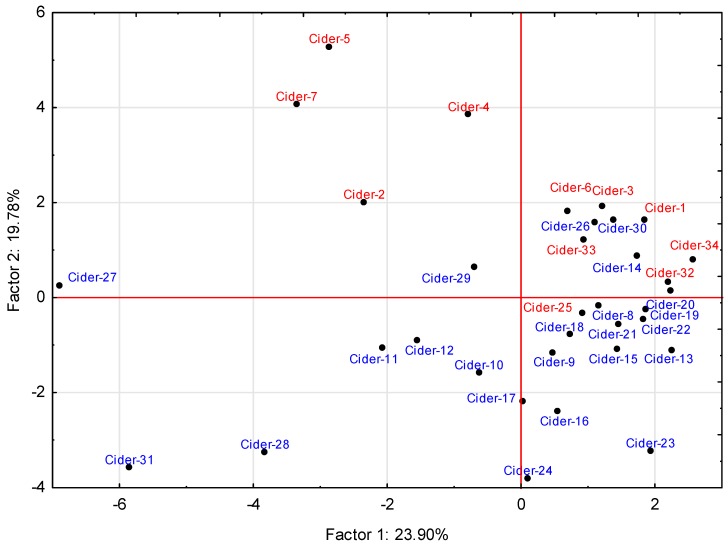
Principal component analysis of the ciders from classic (blue) and intensified production (red) based on Table 2.

**Figure 2 molecules-24-02117-f002:**
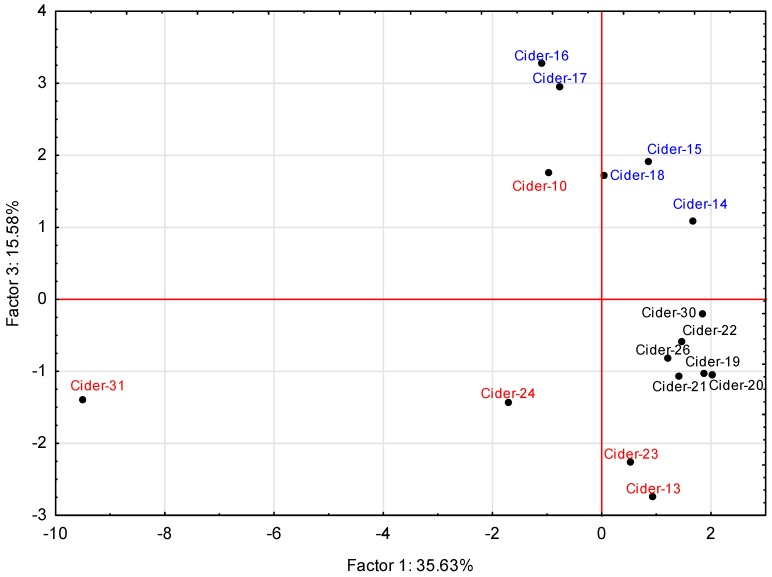
Principal component analysis of ciders fermented by different techniques: exclusively autochthonous culture (red); pure strains of *Saccharomyces cerevisiae* (blue); autochthonous culture with addition of pure strains of *Saccharomyces cerevisiae* (black), based on Table 3.

**Table 1 molecules-24-02117-t001:** Concentrations of volatile compounds in the European ciders analyzed.

Sample Code	Concentration of Volatiles (mg/L)																					
	Ethyl Acetate	Ethyl Butyrate	Ethyl 2-methyl Butyrate	Butyl Acetate	2-Methyl Propanol	Isoamyl Acetate	2-Methyl Butanol	3-Methyl Butanol	Ethyl Hexanoate	Hexyl Acetate	Hexanol	Ethyl Octanoate	Acetic Acid	Butane-2,3-diol	Ethyl Decanoate	Diethyl Succinate	2-Phenyl Ethyl Acetate	Hexanoic Acid	2-Phenyl Ethanol	Ethyl tetradecanoate	Octanoic Acid	Decanoic Acid
Cider-1	0.878	0.003	0.076	0.019	1.576	0.248	1.887	2.914	0.148	0.390	2.302	0.013	1.086	12.123	0.004	0.033	0.007	0.165	3.060	0.001	0.259	0.144
Cider-2	6.792	0.004	0.147	0.004	0.803	2.626	0.755	0.376	0.743	0.378	0.703	0.195	3.697	3.152	0.016	0.131	0.036	2.410	0.960	0.002	3.348	1.530
Cider-3	4.504	0.001	0.139	0.002	1.500	1.746	1.554	1.064	0.054	0.289	0.638	0.010	3.005	4.522	0.015	0.254	n. d.*	0.142	0.396	0.003	0.073	0.025
Cider-4	6.862	0.022	0.366	0.397	1.053	1.237	2.950	1.045	1.063	1.473	1.907	0.020	0.000	10.847	0.011	n. d.*	0.065	0.415	1.222	n. d.*	0.518	0.003
Cider-5	15.194	0.033	0.509	0.649	1.090	2.181	1.679	1.793	1.238	2.713	2.010	0.020	4.638	20.773	0.002	n. d.*	0.142	0.963	1.876	n. d.*	1.192	0.000
Cider-6	8.390	0.009	0.042	0.044	0.872	0.367	1.576	0.063	1.139	0.119	0.416	0.014	2.654	2.784	0.021	n. d.*	0.094	0.854	1.246	n. d.*	0.904	0.044
Cider-7	5.178	0.067	0.155	0.236	0.896	0.601	0.737	0.162	2.611	0.040	0.084	0.219	2.133	n. d.*	0.020	n. d.*	0.025	1.607	0.830	0.001	2.801	1.811
Cider-8	20.150	0.002	0.006	n. d.*	1.079	0.753	2.114	1.589	0.379	0.008	0.902	0.097	13.280	6.656	0.015	0.518	0.010	0.751	1.968	n. d.*	1.491	0.786
Cider-9	12.213	0.008	0.042	0.107	4.677	0.442	7.308	4.003	0.778	0.086	2.567	0.059	9.207	2.279	0.015	0.387	0.006	0.953	4.551	n. d.*	1.248	0.425
Cider-10	8.710	0.002	0.003	0.001	3.351	0.125	4.420	5.816	0.414	0.029	3.541	0.055	4.824	6.112	0.051	0.048	0.067	4.880	14.786	0.001	0.713	1.313
Cider-11	8.302	0.004	0.005	0.297	3.999	1.869	8.058	6.426	0.688	0.644	1.984	0.110	9.863	6.275	0.013	0.163	0.109	1.210	10.458	n. d.*	1.703	0.829
Cider-12	9.646	0.005	0.007	0.499	6.219	1.847	4.829	12.651	0.425	0.785	3.028	0.035	10.139	3.754	0.003	0.121	0.071	1.094	5.106	n. d.*	1.031	0.793
Cider-13	50.342	0.006	0.006	0.030	2.995	0.219	2.877	2.056	0.277	0.097	3.981	0.035	14.010	9.925	0.018	0.193	0.029	1.344	1.707	n. d.*	0.292	0.290
Cider-14	5.086	0.003	0.010	n. d.*	1.505	0.354	2.501	0.534	0.304	0.013	0.235	0.044	0.502	n. d.*	0.006	0.173	0.014	0.624	2.649	n. d.*	0.841	0.138
Cider-15	7.023	0.002	0.005	0.010	8.194	0.212	5.351	4.107	0.231	0.025	0.681	0.036	0.788	n. d.*	0.007	0.275	0.029	0.326	4.158	n. d.*	0.400	0.080
Cider-16	6.442	0.004	0.005	0.004	4.349	0.239	8.030	5.120	0.568	0.025	0.927	0.133	0.624	n. d.*	0.007	2.403	0.060	0.915	11.807	n. d.*	1.061	0.162
Cider-17	5.538	0.002	0.005	0.013	8.432	0.263	4.740	7.193	0.295	0.020	0.609	0.153	0.915	2.852	0.026	0.246	0.058	0.747	13.896	n. d.*	1.142	0.181
Cider-18	4.715	0.002	0.005	0.024	3.180	0.224	5.718	5.179	0.412	0.039	0.672	0.137	0.758	3.898	0.031	0.285	0.031	0.653	5.913	n. d.*	0.748	0.125
Cider-19	10.266	0.003	0.018	0.004	2.180	0.061	2.326	n. d.*	0.197	0.029	4.294	0.058	5.981	16.022	0.073	0.142	0.026	0.485	1.062	n. d.*	0.456	0.534
Cider-20	18.096	0.003	0.030	0.021	2.167	0.062	1.818	n. d.*	0.350	0.021	3.724	0.061	6.119	6.823	0.026	0.212	0.027	0.637	0.731	n. d.*	0.303	0.179
Cider-21	15.937	0.002	0.019	0.003	2.146	0.044	2.915	2.705	0.242	0.015	3.180	0.092	6.807	19.069	0.050	n. d.*	0.006	0.610	1.321	n. d.*	0.897	0.712
Cider-22	8.744	0.004	0.022	0.010	2.911	0.081	5.224	2.384	0.310	0.058	4.650	0.032	2.036	10.871	0.011	0.127	0.011	1.310	0.914	n. d.*	0.366	0.292
Cider-23	31.057	0.002	0.024	0.003	6.457	0.349	5.630	5.223	0.387	0.017	4.320	0.100	18.378	41.361	0.044	0.767	0.008	0.245	3.767	n. d.*	0.520	0.089
Cider-24	21.572	0.003	0.010	0.008	7.197	0.333	8.082	7.334	0.573	0.019	4.196	0.227	11.992	45.493	0.061	0.581	0.009	1.313	4.748	0.001	1.842	0.319
Cider-25	4.251	0.002	0.003	n. d.*	3.705	0.407	3.694	3.111	0.322	0.025	0.501	0.067	1.114	n. d.*	0.018	0.164	0.036	0.681	8.334	n. d.*	0.803	0.340
Cider-26	6.707	0.006	0.049	0.027	1.141	0.296	1.809	n. d.*	0.431	0.605	1.963	0.017	1.833	n. d.*	0.006	n. d.*	0.076	0.280	2.218	n. d.*	0.474	0.495
Cider-27	8.541	0.004	n. d.*	0.057	1.316	2.153	4.528	6.857	1.959	1.314	0.792	0.286	0.381	7.278	0.113	n. d.*	0.197	15.382	7.200	0.001	3.464	1.694
Cider-28	7.491	0.005	0.001	0.013	6.680	0.634	11.352	11.724	1.411	0.343	2.808	0.460	n. d.*	n. d.*	0.102	n. d.*	0.022	2.829	4.083	n. d.*	3.920	0.980
Cider-29	4.738	0.006	0.012	0.027	0.955	0.507	3.798	1.757	1.021	0.219	0.883	0.181	n. d.*	0.927	0.069	n. d.*	0.028	3.261	2.421	0.001	1.898	0.439
Cider-30	5.581	0.005	0.056	0.008	1.089	0.196	1.212	n. d.*	0.276	0.526	1.331	0.026	0.269	n. d.*	0.002	n. d.*	0.041	0.395	3.200	0.001	0.672	0.303
Cider-31	14.687	0.008	0.005	0.197	6.882	1.900	8.133	11.841	0.953	0.509	2.373	0.607	4.422	4.320	0.357	n. d.*	0.079	1.462	6.121	0.002	3.477	0.994
Cider-32	24.239	0.002	0.004	n. d.*	1.073	0.208	1.378	0.564	0.235	0.004	0.336	0.075	5.560	5.933	0.014	0.249	0.014	0.265	3.410	n. d.*	0.434	0.242
Cider-33	3.009	0.010	0.107	n. d.*	1.076	0.410	2.306	1.632	0.361	0.487	3.314	0.035	0.213	n. d.*	0.003	0.099	0.009	0.598	2.357	n. d.*	1.588	0.523
Cider-34	7.651	0.001	0.004	0.001	1.588	0.140	1.781	0.642	0.151	0.006	0.297	0.015	1.626	3.076	0.001	0.309	0.010	0.063	1.865	n. d.*	0.120	0.036

* n. d.—not detected.

**Table 2 molecules-24-02117-t002:** Factor loading of the individual compounds in craft/intensified ciders.

Compounds	Factor 1	Factor 2	Factor 3
Ethyl acetate	0.1929	−0.2871	0.6272
Ethyl butyrate	−0.3748	0.5518	0.1699
Ethyl 2-methyl butanoate	−0.2138	**0.7420 ***	0.4441
Butyl acetate	−0.4613	0.4067	0.5826
2-Methylpropanol	−0.1353	−**0.7864 ***	0.1817
Isoamyl acetate	−**0.7211 ***	0.2663	0.2067
2-Methylbutanol	−0.3542	−**0.7724 ***	0.0937
3-Methylbutanol	−0.5470	−0.6894	0.1476
Ethyl hexanoate	−**0.7743 ***	0.2711	−0.0167
Hexyl acetate	−0.5167	0.5376	0.4625
Hexanol	0.1405	−0.3842	0.6092
Ethyl octanoate	−**0.7346 ***	−0.4686	−0.1391
Acetic acid	0.1652	−0.3695	0.6990
Butane-2,3-diol	0.1088	−0.2486	**0.7352 ***
Ethyl decanoate	−0.5897	−0.4383	−0.0206
Diethyl succinate	0.2157	−0.3754	0.0027
2-Phenylethyl acetate	−0.6702	0.2028	0.1372
Hexanoic acid	−0.6419	−0.0540	−0.1700
2-Phenylethanol	−0.2444	−0.5161	−0.1844
Ethyl tetradecanoate	−0.3062	0.1417	−0.2085
Octanoic acid	−**0.8606 ***	−0.1616	−0.1690
Decanoic acid	−0.6946	−0.0444	−0.2195

* Statistically significant loadings (>0.7000) are shown in bold.

**Table 3 molecules-24-02117-t003:** Factor loading of the individual compounds in ciders fermented by different techniques.

Compounds	Factor 1	Factor 2	Factor 3	Factor 4
Ethyl acetate	−0.0112	0.3976	−0.7095 *	−0.0318
Ethyl butyrate	−0.4807	−0.5699	−0.4527	0.1350
Ethyl 2-methyl butanoate	0.4128	−0.6271	−0.3632	0.1209
Butyl acetate	−**0.8747 ***	−0.3016	−0.2470	0.0976
2-Methylpropanol	−0.5269	0.5379	0.2102	0.2558
Isoamyl acetate	−**0.9196 ***	−0.2094	−0.1727	0.1720
2-Methylbutanol	−0.6976	0.5118	0.2251	0.2068
3-Methylbutanol	−**0.8746 ***	0.3722	0.1742	−0.0042
Ethyl hexanoate	−**0.9127 ***	−0.0729	−0.0601	0.0849
Hexyl acetate	−0.3249	−**0.8384 ***	−0.2469	0.0859
Hexanol	0.0682	0.3553	−**0.7374 ***	−0.3973
Ethyl octanoate	−**0.9593 ***	0.0296	−0.1085	0.1515
Acetic acid	−0.0189	0.5704	−**0.7254 ***	−0.0792
Butane-2,3-diol	−0.0119	0.6782	−0.5551	0.0351
Ethyl decanoate	−**0.9015 ***	−0.0931	−0.2716	−0.0397
Diethyl succinate	−0.0737	0.3759	0.3839	0.3394
2-Phenylethyl acetate	−0.5296	−0.5675	0.3575	−0.2467
Hexanoic acid	−0.2899	0.1588	0.1650	−**0.8843 ***
2-Phenylethanol	−0.3958	0.2003	**0.7164 ***	−0.3428
Ethyl tetradecanoate	0.1141	−0.6584	−0.1195	0.0137
Octanoic acid	−**0.9273 ***	0.0754	−0.0263	0.1076
Decanoic acid	−0.4703	−0.2056	−0.1149	−**0.7974 ***

* Statistically significant loadings (>0.7000) are shown in bold.

**Table 4 molecules-24-02117-t004:** Analyzed ciders and their characteristics based on producers.

Code of Sample	Country of Origin *	Type of Product	ABV **	pH	Condition of Fermentation			
					Autochthonous Culture	Added Yeast without Pasteurization	Pure Yeast Culture	Maturationin the Bottle
Cider-1	CZ	Large-scale	4.0	2.87			X	
Cider-2	CZ	Large-scale	4.0	3.02			X	
Cider-3	CZ	Large-scale	4.0	2.96			X	
Cider-4	CZ	Large-scale	4.5	3.13			X	
Cider-5	CZ	Large-scale	5.0	3.08			X	
Cider-7	CZ	Large-scale	4.0	3.15			X	
Cider-10	CZ	Small-scale	5.5	3.81	X			X
Cider-11	CZ	Small-scale	4.0	3.48		X		
Cider-12	CZ	Small-scale	6.0	3.46		X		
Cider-13	CZ	Small-scale	6.5	3.76	X			X
Cider-27	CZ	Small-scale	5.0	3.64		X		
Cider-28	CZ	Small-scale	5.0	3.46		X		
Cider-29	CZ	Small-scale	5.0	3.90		X		
Cider-31	CZ	Small-scale	4.0	3.66	X			
Cider-8	CZ	Small-scale	4.9	3.70		X		
Cider-9	CZ	Small-scale	4.9	3.55		X		
Cider-23	E	Small-scale	6.0	3.87	X			X
Cider-24	E	Small-scale	6.0	3.78	X			X
Cider-32	EE	Large-scale	4.5	3.56			X	
Cider-19	FR	Small-scale	4.5	3.65	X	X		X
Cider-20	FR	Small-scale	5.5	3.68	X	X		X
Cider-21	FR	Small-scale	4.0	3.73	X	X		X
Cider-22	FR	Small-scale	7.0	3.57	X	X		X
Cider-26	FR	Small-scale	2.0	3.67	X	X		
Cider-30	FR	Small-scale	4.0	3.82	X	X		
Cider-33	GB	Large-scale	4.5	3.12			X	
Cider-34	GB	Large-scale	4.7	3.05			X	
Cider-14	GB	Small-scale	4.0	3.35		X		
Cider-15	GB	Small-scale	5.5	3.24		X		
Cider-16	GB	Small-scale	7.0	3.47		X		
Cider-17	GB	Small-scale	7.0	3.43		X		
Cider-18	GB	Small-scale	8.2	3.47		X		
Cider-25	IE	Large-scale	4.5	3.47			X	
Cider-6	SK	Large-scale	4.0	3.05			X	

* Geographic identification code: CZ—Czech Republic; E—Spain; EE—Estonia; FR—France; GB—Great Britain; IE—Ireland; SK—Slovakia; ** ABV—Alcohol by volume, measured by HPLC.

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
