# Peer review of "An HS-SPME-GC-MS Method for Profiling Volatile Compounds as Related to Technology Used in Cider Production"

_molecules, 2019, doi:10.3390/molecules24112117_

Round 1
Reviewer 1 Report
Revisions
The work studies the volatile composition of cider coming from different European countries and obtained through different production techniques. The data obtained are few (only 22 compounds identified) but well illustrated and explained in the discussions, even the conclusions are exhaustive and in agreement with the data obtained. However, the work presents a major defect of an analytical nature. The experimental part is lacking in detailed descriptions of the SPME technique, and above all not satisfactory in the identification of the compounds.
in any case I consider the work as a whole to be quite satisfactory from the point of view of the food technologies discussed and discussed but mediocre in the analytical part.
revisions to be made are as follows.
Line 87-96. this description of the samples must be moved to the materials and methods section, before preparing the samples (Line 282)
Line 181. the percentage of the factors does not coincide with Figure 1. check the statistical analysis
Line 189. the Factor 3 column can be eliminated since it is not reported in Figure 1 and in the discussion
Line 273. the Factor 3 and Factor 4 columns can be eliminated since they are not shown in Figure 2 and in the discussion
Line 282. the description of the samples used is not reported
Line 290. during the optimization of the extractive method were fibers tested with different stazinary phases?
Line 291. shows the temperatures and times of the individual SPME phases
Line 303. the identification of the compounds was carried out using only the comparison with the NIST database. this cannot be considered sufficient. at least one other method of comparison must be used (eg linear retention rates)
Line 314. 22 compounds identified are few. perhaps with a different fiber and with additional methods of identification a greater number could be obtained.
Author Response
Reviewer 1
Line 87-96. this description of the samples must be moved to the materials and methods section, before preparing the samples (Line 282)
· Description of the samples was moved. Now it is in chapter 3.1.; L: 268.
Line 181. the percentage of the factors does not coincide with Figure 1. check the statistical analysis
· The percentage of the factors was corrected; L: 169-170.
Line 189. the Factor 3 column can be eliminated since it is not reported in Figure 1 and in the discussion
· Factor is now reported in discussion on L: 173.
Line 273. the Factor 3 and Factor 4 columns can be eliminated since they are not shown in Figure 2 and in the discussion
· Factors are now reported in discussion on L: 236-240.
Line 282. the description of the samples used is not reported
· The description of the samples is now in chapter 3.1.
Line 290. during the optimization of the extractive method were fibers tested with different stazinary phases?
· The method was already optimized, as previously published in the reference paper and therefore one specific fiber was chosen for this purpose. It is mentioned in the text; L: 301-305.
Line 291. shows the temperatures and times of the individual SPME phases
· Time and temperature are shown a line L: 305. Isolation was performed under constant temperature.
Line 303. the identification of the compounds was carried out using only the comparison with the NIST database. this cannot be considered sufficient. at least one other method of comparison must be used (eg linear retention rates)
· Identifications were carried out by their specific retention time and also by a comparison with the NIST database. This was corrected in the text; L: 321.
Line 314. 22 compounds identified are few. perhaps with a different fiber and with additional methods of identification a greater number could be obtained.
· Only quantified compounds were mentioned in the text, that was corrected and added; L: 330-331.

Reviewer 2 Report
During the reading of the manuscript, some questions and comments came up and pointed below: 1. Introduction - The authors should more emphasize the novelty of their research. 2. Abstract, Line 24 – please add comma before “and butane-2,3-diol” 3. Abstract, Line 25 – please add “and” before “hexyl acetate” 4. Table 1 – Abbreviations used in the table (CZ, E, EE, FR, etc.) should be explained. Did the alcohol content in individual ciders was taken from the label on the bottle? Please provide this information below the table. 5. Section 2.1. - How did the authors get the technical data of the production of the individual ciders? Did these data were received from producers? 6. General comment for section “Results and discussion” - What criteria did the authors use when choosing volatile compounds used for statistical analysis? What about other volatile compounds? Nothing else was identified? 7. Section 2.3, lines 174-177 – The authors wrote that: “Principal component analysis and factor analysis were applied on two groups of different samples to confirm the assumption that the use of apple juice concentrate or blending of apple wine and apple juice, together with a different approaches to processing, will affect the contents of volatile substances and therefore these samples will have different organoleptic properties.” Did the authors make an organoleptic assessment of the individual ciders? The use of apple juice and a mixture of apple and apple juice will affect the organoleptic properties of ciders. However, not only the volatile compounds investigated in this work affect organoleptic properties. Some of these compounds have low OAV and do not affect the aroma of the product. 8. Section 3.3 - What was the time of desorption of volatile compounds from fiber in the GC inlet?
Author Response
Reviewer 2
1. Introduction - The authors should more emphasize the novelty of their research.
· Authors wrote the introduction as an overview of the current knowledge, and the novelty of this paper is the statistical comparison of the contents of volatiles in ciders with regards to the applied production technology. The novelty of the research was added on lines. L: 82-83.
2. Abstract, Line 24 – please add comma before “and butane-2,3-diol”
· Comma was added; L: 24.
3. Abstract, Line 25 – please add “and” before “hexyl acetate”
· “And” was added; L: 25.
4. Table 1 – Abbreviations used in the table (CZ, E, EE, FR, etc.) should be explained. Did the alcohol content in individual ciders was taken from the label on the bottle? Please provide this information below the table.
· Abbreviations are “commonly used abbreviations” according to ISO 3166-1 for the names of countries, therefore authors didn´t explain them, but for easier understanding of the text explanations were add, now under the table 4 (previously table 1); L: 280-281.
· The Alcohol content was measured in laboratory, description of the method is now in chapter 3.3 and mentioned under the table 4.
5. Section 2.1. - How did the authors get the technical data of the production of the individual ciders? Did these data were received from producers?
· For this paper all the technical data related to the production of the analysed ciders, were obtained from individual producers by personal conversations or received in a paper form.
6. General comment for section “Results and discussion” - What criteria did the authors use when choosing volatile compounds used for statistical analysis? What about other volatile compounds? Nothing else was identified?
· The aim of this research was to analyse the mix of higher alcohols, esters, and short carbon chain fatty acids in one run, under one GC conditions. More than 58 compounds were identified but for quantification, we chose compounds belonging to the main representatives of either pleasant or unpleasant aroma in low alcoholic beverages according to the literature.
· The information about the number of identified compounds was newly added in the text; L: 330-331.
7. Section 2.3, lines 174-177 – The authors wrote that: “Principal component analysis and factor analysis were applied on two groups of different samples to confirm the assumption that the use of apple juice concentrate or blending of apple wine and apple juice, together with a different approaches to processing, will affect the contents of volatile substances and therefore these samples will have different organoleptic properties.” Did the authors make an organoleptic assessment of the individual ciders? The use of apple juice and a mixture of apple and apple juice will affect the organoleptic properties of ciders. However, not only the volatile compounds investigated in this work affect organoleptic properties. Some of these compounds have low OAV and do not affect the aroma of the product.
· The approach to cider production will affect cider composition generally, not only volatiles. Authors were focused on a small specific group of compounds, which are important markers of many pleasant and also unpleasant off-flavours. Authors agree with reviewer, that some of these compounds are below their thresholds for similar beverages, but a synergistic effect of their combination can occur, and therefore the main part of the text is focused on the composition of the group of similar compounds e.g. ratio of the esters and higher alcohols. L: 187-199.
· Organoleptic assessment was made by semi-laic evaluators and differences were found, but only small sensory trials were performed, without sufficient statistical establishment, and evaluators weren´t systematically trained according to methodology for cider sensory evaluation, therefore the results are not appropriate for publishing in the paper, but only as a comment; L 203- 204.
8. Section 3.3 - What was the time of desorption of volatile compounds from fiber in the GC inlet?
· Desorption time was set at 10 min, it was added in the text; L: 314.

Round 2
Reviewer 1 Report
the Authors have made the required changes exhaustively keeping the profile of the work presented.
Therefore, I think that the work can be published in the present form.